# Indirect optical trapping using light driven micro-rotors for reconfigurable hydrodynamic manipulation

Unė G. Būtaitė [1], Graham M. Gibson[1], Ying-Lung D. Ho [2], Mike Taverne [2], Jonathan M. Taylor [1] & David B. Phillips[3]

Optical tweezers are a highly versatile tool for exploration of the mesoscopic world, permitting non-contact manipulation of nanoscale objects. However, direct illumination with intense lasers restricts their use with live biological specimens, and limits the types of materials that can be trapped. Here we demonstrate an indirect optical trapping platform which circumvents these limitations by using hydrodynamic forces to exert nanoscale-precision control over aqueous particles, without directly illuminating them. Our concept is based on optically actuated micro-robotics: closed-loop control enables highly localised flow-fields to be sculpted by precisely piloting the motion of optically-trapped micro-rotors. We demonstrate 2D trapping of absorbing particles which cannot be directly optically trapped, stabilise the position and orientation of yeast cells, and demonstrate independent control over multiple objects simultaneously. Our work expands the capabilities of optical tweezers platforms, and represents a new paradigm for manipulation of aqueous mesoscopic systems.

[1] School of Physics and Astronomy, University of Glasgow, Glasgow G12 8QQ, UK. [2] Department of Electrical and Electronic Engineering, University of Bristol, Bristol BS8 1UB, UK. [3] School of Physics and Astronomy, University of Exeter, Exeter EX4 4QL, UK. Correspondence and requests for materials should be addressed to U.G.B. (email: u.butaite.1@research.gla.ac.uk) or to D.B.P. (email: d.phillips@exeter.ac.uk)

Optically actuated micro-robotics is an emerging field, which exploits the momentum of light to drive micro-mechanical systems, using intelligent control concepts from robotics[1]. Operating within the focal volume of high-magnification microscopes, the over-arching goal of this developing technology is to precisely direct and automate the interaction between specifically designed optically trapped nano-structures and their surroundings, enabling new ways to characterise and explore the mesoscopic world. A key application is the investigation and control of biological systems at the single-cell level and below[2]. In this area, initial steps have led to the development of new methods to characterise the mechanical properties of cells and their response to stimuli[3,4], new scanning imaging techniques to map surface topography with nanoscale resolution[5,6] and new ways to investigate the properties of individual strands of DNA[7,8].

Many of these developments have been made possible through the union of cutting-edge fabrication, manipulation and tracking technologies. Over the past decade, nano-fabrication techniques such as direct laser writing have enabled the construction of near-arbitrarily shaped three-dimensional (3D) micro-structures with nanoscale features[9,10]. Once dispersed in aqueous media, these micro-structures can be dynamically actuated using holographic optical tweezers: multiple focused beams of light that can be independently reconfigured at video rates to trap and manipulate dielectric particles[11,12]. These systems can then be brought to life by handing over control of their motion to a computer. By automatically monitoring the positions of micro-structures in real time[13], and guiding the application of optical forces using feedback[14], they become robotic agents capable of performing tasks well beyond those achievable with manual control[15,16]. They can be programmed to react to their environment on millisecond timescales[17], and their motion can be choreographed with nanoscale finesse[18].

In this work, we employ an optically actuated micro-robotic system to create a new form of near-field hydrodynamic micro-manipulation. Conventional microfluidics-based hydrodynamic tweezers trap and move particles using feedback-controlled flow fields generated by independently adjusting the pressures in a fixed arrangement of converging channels[19–22]. Since these systems manipulate particles using fluid forces alone, their action is independent of particle composition, and minimally damaging to biological specimens. However, during operation, large-scale flow fields are generated throughout the sample, which act indiscriminately on all immersed particles. This typically constrains the application of hydrodynamic tweezers to extremely dilute samples, in order to minimise the risk of bombarding hydrodynamically trapped particles with other objects entrained by the fluid flow. Here, we combine concepts from both optical and hydrodynamic approaches, to create a fully reconfigurable system capable of inducing highly localised flow fields targeted only at specific particles, therefore leaving other objects in the sample largely unperturbed. This system retains the flexibility of optical tweezers, and so can operate anywhere throughout the sample. However, as it relies on hydrodynamic forces, the platform offers a new route to overcome some key limitations of conventional optical tweezers, namely that many types of target particles cannot be directly optically tweezed[23], and that biological systems can be damaged by the high intensities of tightly focused laser beams[24,25]. Using near-field hydrodynamic control, we achieve precise manipulation of one or more free-floating target particles without directly illuminating them: the physical separation between the laser foci and the hydrodynamically trapped objects protects them from photo-damage. The reconfigurable nature of this technique opens up a variety of new experimental paradigms, such as the ability of micro-rotors to rearrange around, and move along with the particles they are steering—akin to hydrodynamic tweezing with a dynamically reconfigurable microfluidic chip.

## Results

**Concept and theory.** Our technique relies on a simple principle: when an optically trapped micro-structure is moved, it displaces pico-litre quantities of the surrounding fluid in a highly predictable manner, exerting well-defined hydrodynamic forces on nearby objects[26]. To harness this concept for controlled micro-manipulation, we have designed mobile optically trappable micro-rotors that, driven by feedback control, allow us to dynamically sculpt flow fields.

Figure 1 illustrates the use of two optically trapped micro-rotors to hydrodynamically control the position of a single target particle in two dimensions. The rotors can be spun on the spot (about their own axis), each entraining the surrounding fluid to generate a hydrodynamic vortex capable of moving a target particle along one dimension in a positive or negative direction. By positioning the rotors orthogonally about the target, each one independently addresses target motion in the $x$ or $y$ dimensions (Fig. 1a). The key to the control system is a real-time feedback loop (Fig. 1b), operating at 200 Hz. In each loop iteration, we track the current location of the target particle, calculate the rotation rates of the rotors needed to push the target towards its required location and move the micro-rotors accordingly. We note that, unlike many hydrodynamic tweezing systems, this control mechanism is quiescent, that is, there is no need to generate flow when the target particle is in its desired location[21].

The micro-rotors are fabricated using direct laser writing[10,27] (see Methods). To create a flow field around each micro-rotor that is independent of its orientation, the design consists of a smooth outer ring, with three internal spokes. Each spoke incorporates a prolate-shaped 'handle' to facilitate precise optical trapping: the elongated shape being chosen to maximise the spatial overlap between the handle and the trapping beam focus (Fig. 1a). Three optical traps per rotor are focused on these handles to give us full translational and rotational control of the micro-rotor[14], with rotary motion being generated by moving the three traps along a circular trajectory.

Our control loop is based on a model of the hydrodynamic interactions between the particles in the system. The evolution of this system is governed by the Langevin equation, which uses Newton's second law to describe the balance of forces on all of the particles in the system[26]. In the low Reynolds number limit, where viscous forces dominate over inertial forces, this becomes:

$$\mathbf{0} = -\zeta\mathbf{v} + \mathbf{f}_{\text{opt}} + \mathbf{f}_{\text{Brn}}. \tag{1}$$

Equation 1 captures the relationship between the hydrodynamic forces $\zeta\mathbf{v}$, the external forces $\mathbf{f}_{\text{opt}}$ representing any optical forces and torques acting on the micro-rotors and $\mathbf{f}_{\text{Brn}}$, the stochastic forces and torques that give rise to Brownian motion. $\zeta$ is the (square) friction tensor coupling the translational and rotational degrees of freedom of all the particles in the system[26,28], and $\mathbf{v}$ is a column vector containing the linear and angular velocities of all particles.

We cannot predict the Brownian component of the forces on particles in the system, $\mathbf{f}_{\text{Brn}}$, but we can use the remaining terms in Eq. (1) to derive (Supplementary Note 1) a single matrix equation encapsulating the relationship between the rotation rate of each rotor and the target particle velocity:

$$\mathbf{v}_{\text{t}} = \mathbf{C}_{\text{tr}}\boldsymbol{\omega}_{\text{r}}, \tag{2}$$

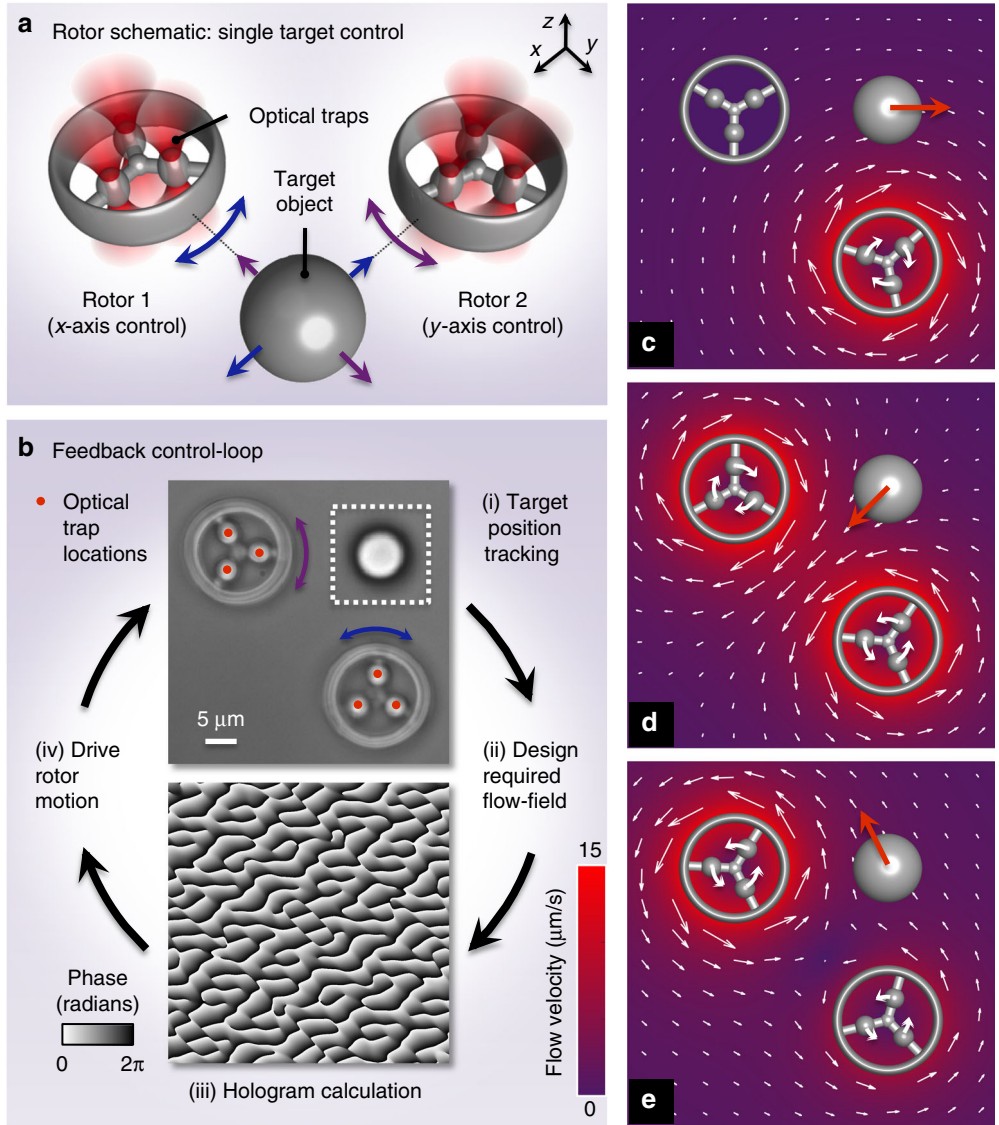

**Fig. 1** Concept. **a** Rotor configuration for single target control. **b** The key steps in the feedback control loop. Upper inset is an optical microscope image of the micro-rotors and a target particle—a silica bead of 5 μm in radius. Lower inset is an example of a phase hologram displayed on the spatial light modulator (SLM) to generate six optical traps. **c**–**e** Modelled flow fields generated around spinning rotors, calculated by numerically integrating Eq. (1) (see Supplementary Note 3 for details). Colour shows magnitude, and arrows show magnitude and direction of flow. The target can be translated in any direction in two dimensions (2D) using a suitable combination of rotation rates and directions of the two rotors

where $\omega_r$ is a column vector representing the rotation rates of the rotors, $C_{tr}$ is a matrix expressing the hydrodynamic coupling between the rotors and the target particle and $v_t$ is the resultant velocity of the target in the $x$–$y$ plane. The components of $C_{tr}$ are derived from the friction tensor, and in each iteration of the feedback loop we solve for the rotor rates $\omega_r$ that achieve the target velocity $v_t$ required to push it towards the desired location. We note that the Langevin equation is nonlinear (in particle position), meaning that the elements of $C_{tr}$ depend upon the configuration of all of the particles in the system. To accommodate this, we must recalculate a new $C_{tr}$ at every iteration of the feedback loop, and ensure that the loop runs fast enough so that there is only a small change in particle configuration from one iteration to the next.

The hydrodynamic interactions of complex-shaped objects such as our rotors can be modelled by representing them as rigid shells of small hydrodynamically coupled beads[29]. However, to enable real-time flow-field calculation, we reduce the complexity

of the model by treating each rotor as a single spherical particle, which gives a good approximation to the flow field generated by the wheel-shaped micro-rotors. Details of the algorithm and equations can be found in Supplementary Notes 1–4, and we will see later that our mathematical framework is quite general and naturally extends to systems with arbitrarily positioned rotors, more rotors than the number of target degrees of freedom and arbitrary rotor geometries.

**Hydrodynamic clamping of a single particle**. To characterise the precision of the optically actuated hydrodynamic trapping system, we first task the feedback loop to maintain a target object at a fixed location, thus suppressing its Brownian motion in directions parallel to the focal plane of the microscope (the $x$–$y$ plane). Figure 2, and Supplementary Video 1, show that we can 'clamp' (i.e. limit) the Brownian fluctuations of a 5 μm radius silica bead in water to a standard deviation of 79 nm. We note that in all of

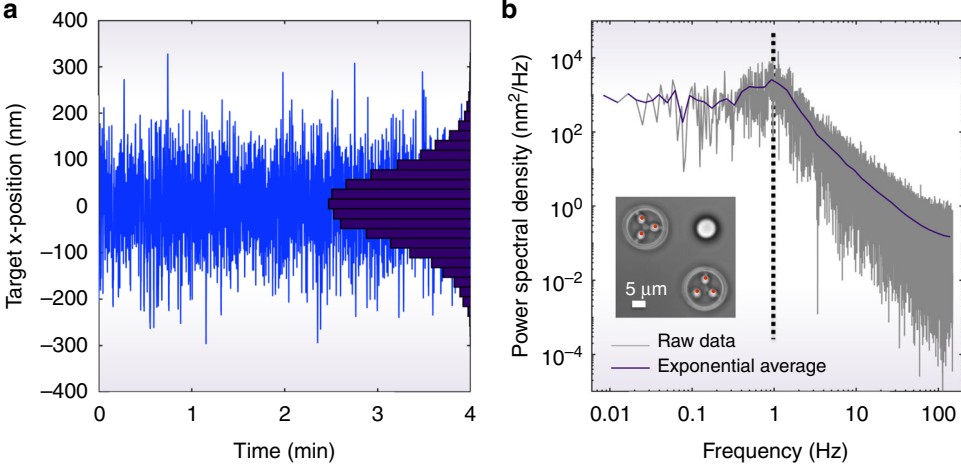

**Fig. 2** Experimental clamping. **a** Measured $x$-position of a hydrodynamically position clamped target particle—in this case a 5 μm radius silica bead target particle (optical image of the experimental set-up is shown in **b**). The target is held in water for a period of 4 min, during which time the standard deviation of the target's motion is 79 nm. Also see Supplementary Video 1. The histogram inset on the right shows the relative occupancy of the positions explored by the particle around its equilibrium position (arbitrary units). **b** Power spectral density of target motion. The corner frequency is ~1 Hz

the experiments that follow, the target particles are sedimented to the bottom surface of the sample, eliminating the need to hydrodynamically control their motion in the third spatial dimension ($z$).

Modelling the hydrodynamic restoring force $f_h$ acting on the target particle as being linearly proportional to its displacement $x$, we can calculate the target's hydrodynamic clamping stiffness $\kappa_x$, such that $f_h = -\kappa_x x$. According to the Equipartition Theorem, $\kappa_x = k_B T / \langle x^2 \rangle$, where $k_B$ is Boltzmann's constant, $T$ is temperature and $\langle x^2 \rangle$ is the clamped target particle's variance[30]. Here we find that $\kappa_x \sim 6 \times 10^{-7}\,\mathrm{Nm}^{-1}$, comparable to that of a weak optical trap.

The degree to which Brownian fluctuations can be suppressed by our system depends on a number of parameters, including the optical trapping stiffness, geometry and hydrodynamic friction of the rotors (which determines their frequency response and maximum rotation rate); the feedback rate (200 Hz) and delay time (i.e. the time from measuring the target's position to the movement of the optical traps: here ~15 ms); and the size of the target particle. In particular, we note that smaller target particles can diffuse further within the feedback delay time, and so are less tightly clamped. For example, a 2.5 μm radius silica bead in water can be clamped in our prototype system with a standard deviation of 186 nm (see Supplementary Fig. 1).

**Hydrodynamic micro-manipulation strategies**. In addition to clamping a target particle in one location, we can use our system to translate the target along a prescribed trajectory through the sample, while simultaneously suppressing Brownian motion to minimise deviations from this path. There are three distinct ways this translation can be achieved, as follows.

Firstly, we can propel the target across the field of view, while keeping the centre-of-mass position of the micro-rotors static. Figure 3a and Supplementary Video 2 illustrate this strategy, showing the motion of the target as it is driven between two points situated on either side of a pair of rotors. This allows us to characterise the target's translational velocity as a function of its distance from the rotors. As expected from Eq. (1), we observe the velocity of the target to be highest when it is in close proximity to the rotors, scaling as a function of $1/d^2$, where $d$ is the target–rotor separation (see Supplementary Note 5 and Supplementary Fig. 2). However, we note that there is a trade-off here: the closer the target is to the rotors, the greater exposure it has to

the high-angled rays of the high numerical aperture (NA) optical traps. In this experiment, the peak target velocity was ~11 μm s$^{-1}$ at an edge-to-edge target–rotor distance of ~0.5 μm. In contrast to the clamping efficiency, the speed at which it is possible to translate an object with the system is nominally independent of the object's size (see Supplementary Note 5). Figure 3b demonstrates the stepping accuracy of our system—showing position histograms as the target is moved in a series of 80 nm steps.

Secondly, an advantage of our system over other forms of flow control is that the micro-rotors are mobile in nature. We are therefore at liberty to reconfigure their positions within the field of view in order to most efficiently achieve the task in hand. For example, exploiting this capability allows the rotors to automatically follow the target as it is translated. Supplementary Video 3 demonstrates this strategy.

Thirdly, by moving the microscope stage, the sample is translated past the field of view of the camera. Meanwhile, the rotors remain optically trapped in the frame of reference of the camera, that is, they are dragged relative to the fluid. In this scenario, the feedback system automatically spins up the rotors, creating a flow field that counters the stage-induced movement of the target, and maintains it at its required location in the frame of reference of the camera. The net result is that the target is propelled relative to the surrounding sample region. Figure 3c–h and Supplementary Video 4 demonstrate this approach.

Note that in this last example we have in fact used three rotors to control two degrees of freedom of target motion. This renders Eq. (2) underdetermined, and thus there are infinitely many possible solutions describing rotor movement that will achieve the desired target motion. However, as the unknown parameter space is one dimensional, it is straightforward to find an optimal solution using a simple iterative search algorithm that runs in real time. Here we define the optimal solution to be the one in which the maximum rotor angular velocity is the lowest—so as to most efficiently generate the desired flow field (see Supplementary Note 4). Throughout the experiment shown in Fig. 3c–h, we once again make use of the mobile nature of the rotors by programming them to dynamically reorient about the target to a configuration optimised for the current direction of travel. Unlike other forms of flow control, which operate only within predefined regions of the sample[19], this third approach gives us

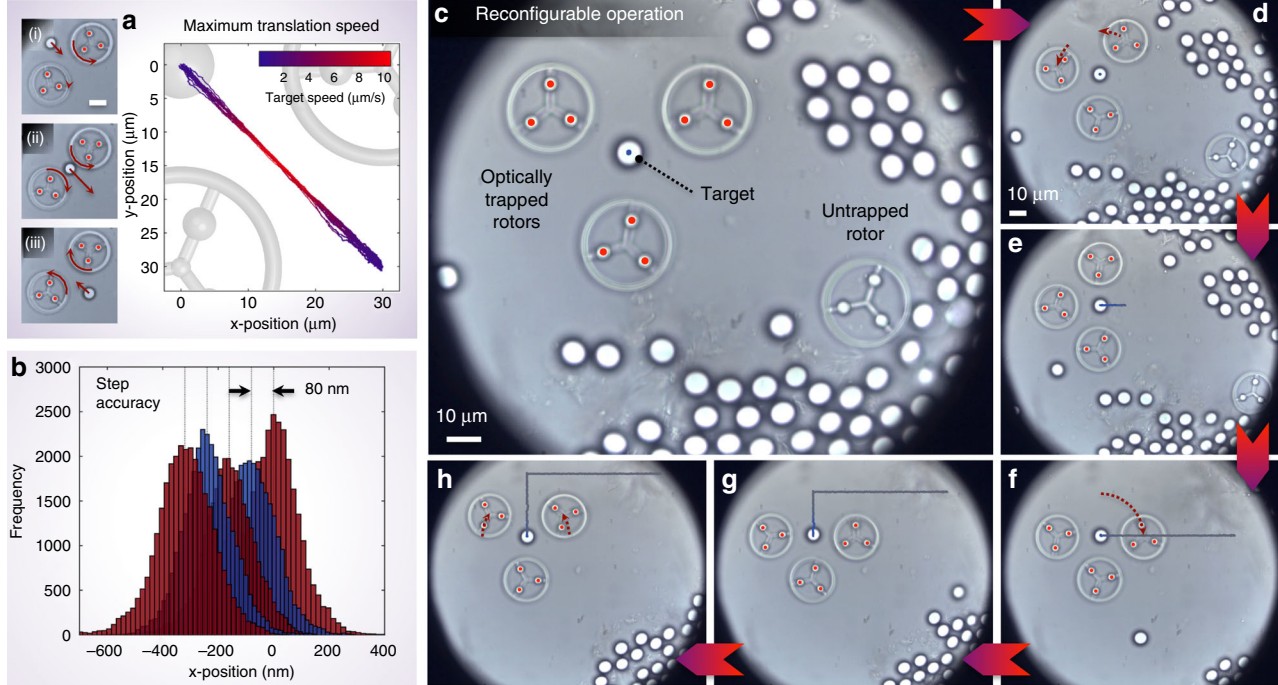

**Fig. 3** Micro-manipulation performance. **a** Maximum translation speed: graph showing the trajectory of a target as it is propelled back and forth between two rotors. Line colour shows target speed. Insets (i)–(iii) show snapshots of the motion from Supplementary Video 2. **b** Stepping accuracy: histograms of target position as it is intermittently moved in 80 nm steps. Minimum step size is ultimately limited only by target tracking accuracy and system drift. **c–h** Reconfigurable operation in crowded environments throughout the sample: six frames from Supplementary Video 4, showing a target silica bead being stabilised and translated through the sample, without disturbing a cluster of beads nearby. Trajectory of the target through the sample is shown as a trailing blue-grey line. **e–h** share the same scale bar as **d**

access to any areas of the sample that can be reached by translating the stage in 2D.

**Constellation rotors**. Fabrication of bespoke micro-rotors using direct laser writing requires access to specialised equipment. Therefore, to circumvent this requirement, next we investigate the performance of our hydrodynamic tweezing system using a simplified rotor structure: a constellation of three independent silica beads (which are readily commercially available), held in a rigid triangular configuration by three optical traps, as shown in Fig. 4.

These spinning bead constellations act as fluid impellers in the same way as the bespoke laser-written micro-rotors[31]. However, rather than generating a uniform hydrodynamic vortex when spun, the rotational symmetry of the fluid flow around each bead constellation is broken, and therefore its effect on a target particle will be orientation-dependent (Fig. 4a, b). Despite this additional complexity, the complete orientation-dependent hydrodynamic coupling between the bead constellations and target can still be captured in the coupling matrix $\mathbf{C}_{tr}$ (derivation in Supplementary Note 1). We can therefore use exactly the same feedback loop as described above to achieve hydrodynamic control using bead constellation rotors.

In order to benchmark the performance of the bead constellation rotor approach, we once again hydrodynamically clamp a 5 μm radius silica target, as shown in Fig. 4c, d and Supplementary Video 5. In this case, the system suppresses the Brownian fluctuations of the target to a standard deviation of $\sigma_{\text{mean}} = 89$ nm. We attribute the marginally reduced performance in comparison with laser-written rotors (~10%) to the fact that accounting for the coupling between the constellation rotors results in an orientation-dependent clamping efficiency, and that due to their size the constellation beads are less tightly trapped

than the laser-written micro-rotors, reducing their responsiveness to higher-frequency optical trap movements (see Discussion).

Next, we challenge the feedback loop with a more complicated task: movement of the target along a complex prescribed trajectory: a $17 \times 8$ μm University of Glasgow logo (Fig. 4e). Here, we maintain the centre of the constellation rotors at a distance of 22 μm from the target throughout the trajectory, and we see that the target is successfully driven along the complex path over a larger area of the field of view.

To confirm our system's independence of target material, Fig. 4f, g and Supplementary Video 6 show hydrodynamic control over an optically untrappable object for an extended period of time. Here we use constellation rotors to clamp an irregularly shaped chromium fragment, roughly three microns across, for half an hour with $\sigma_{\text{mean}} = 482$ nm. The clamping efficiency is lower in this case due to the fragment's smaller size and its non-spherical shape resulting in a reduced tracking accuracy. Mesoscopic metallic objects such as this cannot be directly optically trapped due to their high reflectivity and rapid heating (via absorption of laser light), but our indirect hydrodynamic trapping is effective regardless of the optical properties of the target particle. We note that optical bottle beams have been used to trap absorbing particles and particles of lower index than their surroundings[32], and manipulation of metallic and silica particles has been achieved using direct contact with clusters of optically trapped beads[33,34]. Here our experiment demonstrates a new minimally invasive trapping method, avoiding heating of absorbing particles by keeping the optical traps separated by ~20 μm from them.

**Orientation control**. Figure 4i and Supplementary Video 7 demonstrate the biological compatibility of our platform by

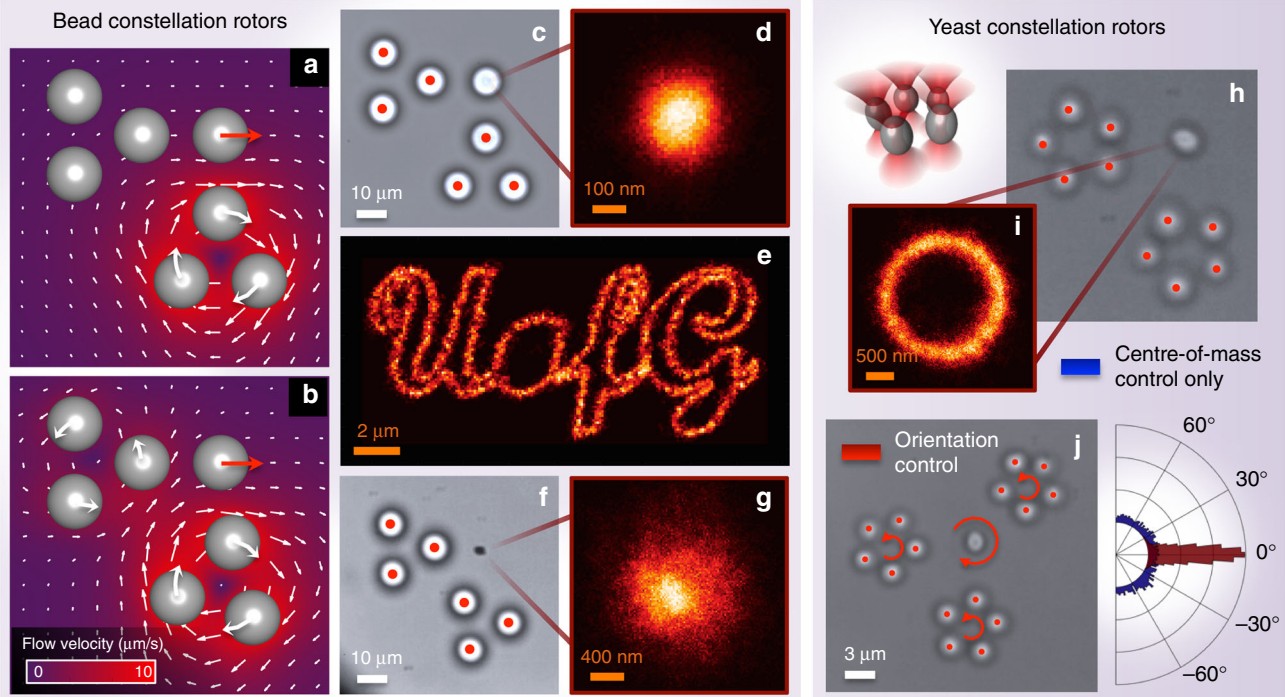

**Fig. 4** Constellation rotors. **a**, **b** Modelled flow fields generated by three-bead (each of 5 μm radius) constellation rotors. In **a** translation of the target in the positive *x*-direction is achieved by spinning just the lower rotor clockwise. In **b** the orientation of the lower rotor has changed. In this case, translation of the target in the positive *x*-direction requires actuation of both rotors. **c** Clamping of a 5 μm radius silica bead using constellation rotors. **d** shows a 2D occupancy histogram of target positions during a 5 min period. **e** Translation of a 5 μm radius silica bead along a complex trajectory tracing out the University of Glasgow logo, using constellation rotors. **f** Clamping of a chromium fragment. **g** shows a two-dimensional (2D) occupancy histogram of the trapped chromium fragment over a 30 min period. Yeast constellation rotors: **h** constellation rotors can also be assembled from yeast particles—even if their precise size is variable and unknown, as illustrated in the schematic top left. The small size (~2–3 μm in diameter) of the yeast particles enables arrangement of five optically trapped yeast in a ring. **i** Micro-manipulation of an individual yeast particle along a circular trajectory of 1 μm radius over a period of 30 min, using yeast constellation rotors. Here while the centre of mass is controlled, the orientation of the target yeast is not. **j** Introduction of a third rotor enables control of target orientation as well as position: for example, spinning all rotors anti-clockwise rotates the target clockwise. To the right is a histogram of the orientations explored by the target in **h** (blue) and **j** (red)

hydrodynamically clamping a yeast cell. Here we slowly move the target yeast cell in a circle of 1 μm radius for half an hour. In fact, in this experiment the rotors themselves are formed from constellations of five 'sacrificial' optically trapped yeast cells. Despite the natural variability in the size and shape of these cells, we are able to treat each one as a spherical particle of radius 1.5 μm, which demonstrates that the high feedback rate of the system can correct any small mis-estimation of the hydrodynamic coupling in the system.

We note that yeast cells are more absorbing of infrared laser light than synthetic rotors, causing some localised heating. We believe this is the cause of a weak convection current that can be observed to gradually draw in surrounding yeast cells during the experiment shown in Supplementary Video 7, an effect not observed when using synthetic rotors. However, as can be seen, these intrusions are directed towards the rotors themselves, and so do not significantly perturb the hydrodynamically clamped yeast particle. This absorption, which is potentially damaging to the yeast cells[35], also serves to highlight the need for the development of new indirect micro-manipulation techniques.

In this experiment, while we stabilise the centre of mass of the prolate yeast cell, its orientation is uncontrolled. Therefore, next we modify the control system to track both the 2D position and orientation of the target cell[36], and introduce a third yeast constellation rotor to stabilise both the target's position and in-plane orientation (Fig. 4j and Supplementary Video 8). This increase in dimensionality is straightforward to implement within

our control architecture: Eq. (2) can be expanded to capture the hydrodynamic coupling between three rotors and target motion in three dimensions (two positions and one orientation). The use of yeast constellation rotors demonstrates the ability to manipulate free-floating biological specimens without introducing any foreign material into the sample in the form of beads or laser-written rotors.

**Independent control of multiple particles.** Finally, we consider the independent hydrodynamic manipulation of multiple target particles. This requires the generation of flow fields with higher levels of complexity—specifically, those incorporating stagnation points. Once again, we expand Eq. (2) to include more dimensions. At first it appears that we need to only include as many rotors as the total number of target degrees of freedom—for example, four rotors to independently control the motion of two targets in 2D. However, we find that in some cases the equations become degenerate and no solution exists. For example, Fig. 5e (inset) illustrates a configuration for which no combination of angular velocities of the rotors can push the two targets together. To overcome this limitation, we break the inherent symmetry by introducing an additional rotor (Fig. 5a), and hence one further degree of freedom, into the system. As seen earlier, Eq. (2) then becomes underdetermined, and we solve in the same way as described above (see also Supplementary Note 4).

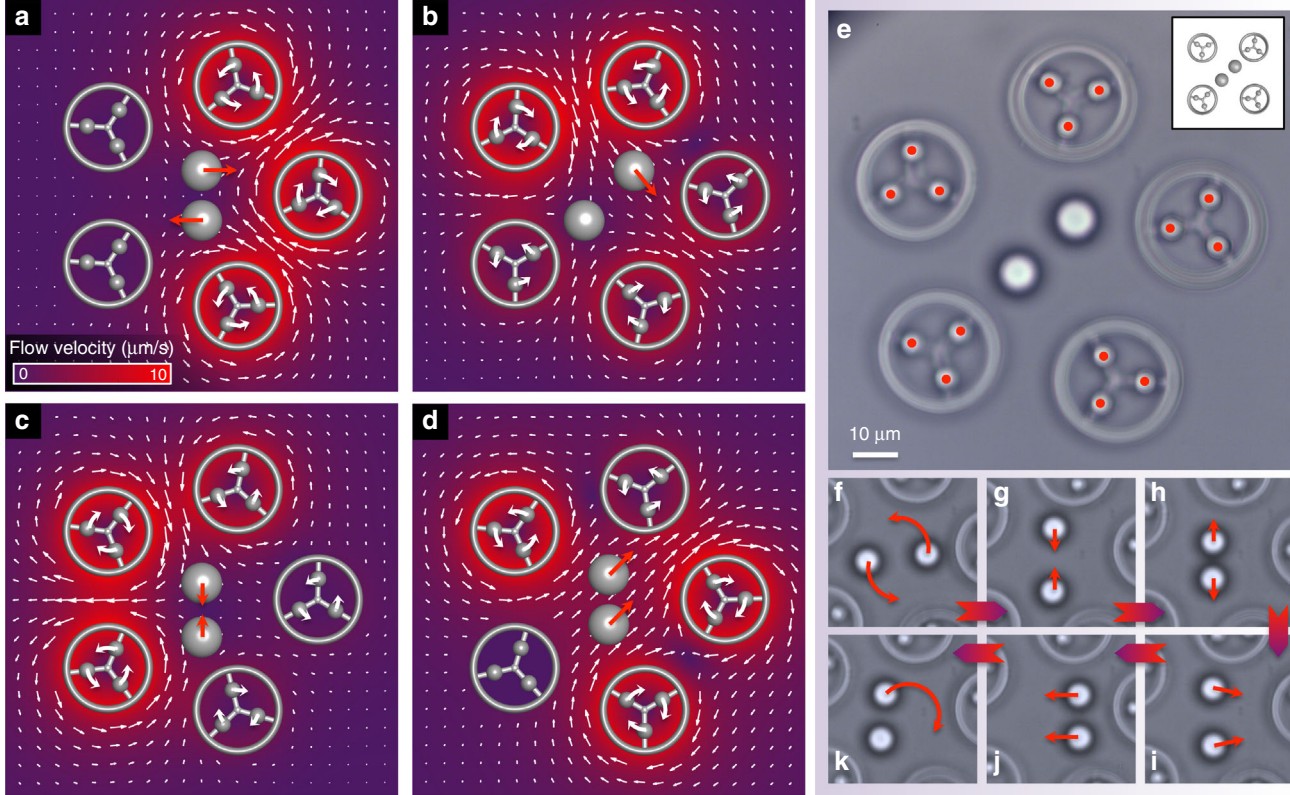

**Fig. 5** Control of multiple particles simultaneously. Five micro-rotors arranged in a ring can be used to control four target degrees of freedom in two dimensions (2D). **a**–**d** Simulations of rotor actuation to push the targets in a variety of directions. In order to move the targets in opposite directions, **a**, **c** require the generation of a flow stagnation point between the targets. In **b**, the lower left target is held stationary by creating a flow stagnation point at its location, while the upper right target is translated. In **d** no flow stagnation point is necessary as both targets are moved in the same direction. Each case shows the optimum combination of rotor motions to achieve the required target translations. We note that with this many degrees of freedom, it is sometimes not an intuitive result. **e** Optical image of the experimental five rotor set-up. Inset shows a schematic of a four rotor set-up. In this case, there is no way the rotors can be spun to push the targets together. **f**–**k** Snapshots from Supplementary Video 9 zoomed in on the central region, showing the targets executing a variety of independent motions

To test the multi-target control capabilities of the system, we first demonstrate independent position clamping of two targets held 14 μm apart (see Supplementary Note 6 and Supplementary Fig. 3). Next, we drive the two targets along independently prescribed trajectories (Fig. 5f–i and Supplementary Video 9). We orchestrate a variety of different motions: the two beads moving anti-clockwise on the same ring, being pushed together and pulled apart, following concentric trajectories in opposite directions, moving the centre of mass with fixed relative particle separation and, finally, holding one target stationary while the other one is tracing out an arc. Supplementary Fig. 4 shows a detailed comparison of the required versus actual trajectories taken by each bead. The level of control we demonstrate over two target particles in very close proximity, using fluid forces alone, suggests future applications of our system to the investigation of femto-Newton scale cell–cell interactions in a minimally invasive manner.

## Discussion

In any hydrodynamic manipulation system, the physical extent of the control flow field is governed by the distance between the flow-actuating elements and the target objects. Conventional hydrodynamic tweezing systems rely on pressure controllers or syringe pumps: since these elements are typically external to the microfluidic control area, flows of macro-scale extent must necessarily be created[19–22]. In this work, we have essentially shrunken the flow-actuating elements to the micro-scale, and brought them in close proximity to (i.e. within ~μm range) the target particles. We have demonstrated that this approach provides levels of performance approaching that of the state of the art in microfluidics-based hydrodynamic tweezers, both in terms of target translation velocity and clamping stiffness. At the same time, our platform demonstrates promising new capabilities in terms of the generation of localised flow fields that can be sculpted to exert nanoscale-precision position control over specific objects, while leaving the surrounding particles largely unperturbed. In contrast to conventional hydrodynamic tweezers, this enables operation even in relatively crowded samples. The near-field nature of our system also facilitates accurate manipulation of multiple particles in very close proximity. In addition, our mobile flow controllers can be reconfigured to operate anywhere within a sample, and can even move with target objects as they are translated, in order to maintain them under tight control.

So what are the factors limiting the performance of our system? The most obvious factor affecting the target clamping stiffness is the feedback delay time, $\tau_d$, during which the target is able to diffuse away from its registered position. $\tau_d$ is determined by the computational time required for image registration, flow calculation and hologram generation, as well as the liquid crystal response time. Together these account for a delay of ~15 ms. To a lesser extent, the feedback rate of 200 Hz (i.e. optical traps repositioned every 5 ms) also limits the target clamping stiffness.

An important additional factor that indirectly affects the target's hydrodynamic clamping stiffness is the optical trapping stiffness with which the rotors are held. In our proof-of-principle experiments, this is in fact the dominant limitation on system performance. The micro-rotors exhibit a roll-off in their response to optical trap motion at high frequencies, resulting in the generation of flow fields with a dynamic modulation that is effectively low-pass filtered in comparison to the optical trap motion. The frequency at which this roll-off occurs is proportional to optical trapping stiffness[30]. Therefore, using micro-rotors that are optically trapped more stiffly (e.g. with higher laser power) will increase the corner frequency of the residual Brownian motion of the hydrodynamically clamped objects. For our experiments, the laser-written rotors were designed with small handles to increase their trapping stiffness and responsiveness. In future, this can be further improved with a combination of optimising the optical interaction between the rotors and the optical traps, and potentially compensating for the known frequency response of the trapped rotors in the control system.

What are the limits on the number of degrees of freedom that may be controlled with our system? The number of required actuators scales linearly with the number of target degrees of freedom we wish to control. As this number increases, we observe a decrease in the clamping stiffness in each dimension. This reduction is in part due to the laser power being shared between more rotors (with an accompanying reduction in spatial light modulator (SLM) diffraction efficiency). However, there is a more fundamental factor common to any hydrodynamics-based control system: for some classes of motion, flows must be generated that contain vortices and stagnation points in close proximity to the targets. For example, in Fig. 4j, orientation control is achieved by creating a hydrodynamic vortex centred near the target yeast; in Fig. 5a stagnation point between the targets must be created to push them together or pull them apart. The presence of these zero-flow points inevitably reduces the magnitude of the target velocities that can be induced.

Our near-field hydrodynamic trapping system exploits the transfer of linear optical momentum from optical tweezers to power our micro-rotors. This leads to the exertion of pico-Newton scale forces, and atto-Newton-metre scale torques. We note that a variety of other rotor types have been reported in the literature in passive flow-generation applications, and these could also potentially be used with our closed-loop feedback platform. These may exploit the transfer of linear momentum[37,38], spin angular momentum[39,40] or orbital angular momentum[41] of light. Alternatively, magnetically driven rotating particles have been used to generate hydrodynamic vortices to trap and move inert particles and cells[42,43], and optoelectronic tweezers are in principle able to apply nano-Newton-scale reconfigurable 2D optical force fields[44]. More generally, active control over colloidal systems has been demonstrated using a range of other physical forces. For example, electrophoretic traps exploit feedback rates of up to ~100 kHz to suppress the Brownian motion of individual fluorescent molecules, but expose particles to strong electric fields, and are limited to operation in predefined target areas of thin samples[45,46].

There are a variety of ways the technology demonstrated here could be further developed. Optical tweezers are well known for their ultra-sensitive pico-Newton scale force transduction capabilities[30]. This sensitivity arises from the relatively weak nature of optical forces: objects in a weak force field move a relatively large distance upon application of a small external force. Our hydrodynamic tweezing platform, by virtue of its even weaker nature (here exerting forces roughly an order of magnitude weaker than optical tweezers), has the potential to be deployed as a force transducer with even greater sensitivity. This could be achieved by tracking the motion of all target and actuator particles, and deducing any unknown external femto-Newton scale forces felt by the target particles by taking account of all hydrodynamic interactions in the system.

Our near-field hydrodynamic manipulation technique may potentially be extended to the third translational dimension, perpendicular to the microscope focal plane. 3D hydrodynamic manipulation of neutrally buoyant target particles (that do not sediment) would be possible by employing optically trapped actuators that can be rotated about arbitrary axes[47,48], in conjunction with 3D imaging techniques[49]. Generating lift on sedimenting objects would, however, require considerably more powerful flow controllers[44] in order to overcome gravitational forces. For example, far from the substrate the terminal sedimentation velocity $v_s$ of a silica particle of radius $r = 5\,\mu m$ in water is ~60 $\mu m\,s^{-1}$ (estimated by finding the velocity at which gravitational and viscous forces balance[50]). However, we note that $v_s \propto r^2$, and so lift may be more readily generated on smaller particles. Despite these challenges, choreographing the motion of clusters of mobile flow micro-actuators has the potential to yield minimally invasive control over any target degrees of freedom that can be measured. We envisage future systems capable of exerting full 3D control over all translational and rotational degrees of freedom of mobile cells, using the minimally invasive forces of water alone.

In summary, our near-field hydrodynamic micro-manipulation system represents a step forward in the development of sophisticated optically actuated micro-robotic systems. We have demonstrated this approach with a variety of different actuator types, but the concept of feedback-based control underpinned by precise hydrodynamic modelling is readily extensible to other actuator platforms. Our technique provides a complementary approach to conventional microfluidics-based hydrodynamic micro-manipulation and electrophoretic trapping, and expands the capabilities of holographic optical tweezers. We have shown a new route to the simultaneous control of multiple microscopic objects of arbitrary material and shape, and a new minimally invasive technique to stabilise and study biological systems while avoiding their irradiation with intense laser light.

## Methods

**Brightfield imaging and optical tracking**. Our hydrodynamic tweezing platform is based on a custom inverted microscope equipped with a holographic optical tweezers arm[51]. A detailed schematic of the optical set-up is shown in Supplementary Fig. 5 (with a detailed explanation in Supplementary Note 7). The sample is illuminated from above using Kohler illumination from a halogen illumination module (Zeiss Axiovert: 100 W) fitted with a 0.55 NA condenser. Light from the sample is collected by the high NA oil immersion objective lens (Nikon Plan Fluor: ×100; 1.3 NA) and imaged onto two cameras with different fields of view and frame rates. Camera 1 (GigE Vision, Teledyne DALSA Genie: HM1024) images the sample at a frame rate of ~400 Hz. The camera's region of interest is cropped to a ~20 × 20 μm field of view around the target particle(s). If necessary, this region of interest is adaptively moved to keep the target centred. Images from Camera 1 are used for high-speed real-time target registration to drive the hydrodynamic feedback system. Camera 2 (USB 3.0, JAI GO: JAI GO-5000M-USB) images the entire field of view of the microscope (a circular aperture of ~150 μm in diameter) at video rates, enabling the recording of full-field movies of the experimental dynamics.

Object tracking is performed in two ways: 2D position coordinates are determined using an image centre-of-mass tracking C library (developed by R. Bowman[52]), called from LabVIEW; or 2D position and orientation coordinates are determined using the custom-written LabVIEW software. This second method generates a binary image by applying a threshold operation to the image. 2D positions are then computed from the centre of mass of this binary particle image. The orientation is then recovered by computing the eigenvectors of the moment of inertia tensor of the binary particle, which point along the major and minor axis of the shape[36].

**Holographic optical tweezers**. A beam from a continuous-wave diode-pumped solid-state laser (Laser Quantum: VentusIR, 3 W), with a wavelength of 1064 nm, is expanded to slightly overfill an SLM (Boulder Nonlinear Systems: XY-series,

512 × 512 resolution). The maximum update rate of the SLM is 204 Hz, and it incorporates a dielectric reflective backplane for high-efficiency operation at 1064 nm. The SLM plane is re-imaged to the back aperture of the microscope objective lens using 4-f relay optics. The objective then focuses the light diffracted from the SLM into the sample to form the optical traps. The overall optical efficiency of the system is ~40%.

The phase holograms that are applied to the SLM are designed to diffract incident light into multiple optical traps in the required positions in the sample. These holograms are calculated using the 'gratings and lenses' algorithm[53,54], which works by back-propagating a monochromatic beam from each required trap location to the SLM plane, and applying a phase pattern, corresponding to the argument of the resulting complex interference pattern, to the SLM. As the SLM is a phase-only modulator, lossy pseudo-amplitude modulation can also be applied to the hologram pattern by varying the grating contrast on the SLM as a function of the required amplitude[55,56]. This redirects unwanted light towards the zero order of the diffraction pattern, minimising the formation of ghost traps (i.e. additional traps appearing in unwanted locations in the sample due to the phase-only nature of the SLM modulation).

The gratings and lenses algorithm is implemented in the OpenGL language on a graphics processor, the inherent parallelisation making it computationally fast[54]. However, even with the application of amplitude modulation, the algorithm does still suffer from lower-power unwanted ghost traps in the sample when working over larger fields of view. Scattering from the liquid crystal in the SLM also causes low-level background light to be spread over the sample plane. Therefore, to ensure the target volume was free from randomly scattered light and ghost traps, in some experiments a beam stop (an opaque spot on a glass slide, aligned using a manual $x$–$y$ translation stage) was placed in a conjugate image plane of the sample target region, blocking all laser light from reaching this location (see Supplementary Fig. 6).

**Control software.** The software driving our hydrodynamic manipulation platform is based on a modified version of a custom-written LabVIEW interface: *Red Tweezers*[54]. This enables interactive manual control of optical trap positions by pointing and clicking the mouse directly on a real-time video of the sample, which is used at the start of experiments to manually trap the micro-rotors before their control is handed over to the computer. For this work, we modified *Red Tweezers* to incorporate our closed-loop hydrodynamic feedback control system to automatically position optical traps. Target objects are tracked in real time (see above), and this information is fed into our control software, which uses Eq. (2) to compute the required rotational velocities for the rotors that will generate hydrodynamic flows to transport the target(s) back to its desired location. The full equations used to solve the hydrodynamic coupling problem can be found in Supplementary Notes 1–4. Based on the calculated rotational velocities that are required, we update the required positions of the optical tweezer foci that should be applied at the next SLM refresh cycle.

If one or more targets are to be driven along a controlled trajectory, the required location of the target is updated at each time-step. In our experiments, the system waits for the target to catch up to within a specified distance (e.g. 1 μm) from its required location before further advancing the required location. This behaviour leads to target objects being translated faster when the required flow fields can be generated more efficiently. However, if necessary then constant translation velocity can also be enforced within the limits of the system.

In some experiments the rotors reconfigure about the target (e.g. Figure 3c). In this case the centre of mass of each rotor follows predefined circular arcs around the target. During these manoeuvres the feedback loop remains on: it is constantly updated with current target and optical trap positions, and so even while the rotors are translated, the control system is able to continue spinning them accordingly to maintain hydrodynamic control over the target.

**Sample preparation.** Micro-rotors are laser printed using a commercially available 3D laser lithography system (Nanoscribe Photonic Professional). Direct laser writing relies on the local solidification of a photoresist (Nanoscribe IP-L) at the focus of a laser beam (a wavelength of 780 nm was used in this case) using a two-photon polymerisation process. By sweeping the beam through the photoresist, optical-quality 3D structures are reproducibly 'drawn' with feature sizes down to 100 nm. Once an array of ~100 micro-rotors has been created, the unpolymerised photoresist is washed away, leaving the array anchored to a glass substrate. The array is then immersed in a droplet of 1% water-Tween-20 (Sigma) solution (to prevent sticking). Viewed under a ×5 magnification microscope, a thin copper wire mounted on a manually controlled XYZ translation stage was used to gently nudge the micro-rotors off the surface into the water solution. Using a pipette, the micro-rotor suspension was transferred into a clean glass sample cell, consisting of a glass slide and coverslip separated by ~150 μm using coverslip spacers. Following this, the rest of the sample cell was filled with a solution of target particles; for example, a weak concentration of a 5 and/or 2.5 μm radius silica beads (Bangs Laboratories), chromium powder (GoodFellow) or fast action dried bakers yeast (Sainsburys). Prior to being filled, all sample cells were cleaned in an ultrasonic bath (Grant) filled with a mixture of sterile purified water (Calbiochem) and ethanol absolute (VWR). To seal the samples, we used either Norland Optical Adhesive 68 and 81 (Norland Products) or transparent nail varnish.

## Data availability

All data needed to evaluate the conclusions in the paper are present in the paper and/or the Supplementary Materials. The raw data for this article can be found in an open-access repository at http://researchdata.gla.ac.uk/id/eprint/714. These data support the following figures: Figs. 2, 3a, b, 4d, e, g, i, j and Supplementary Figs 1–4.

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

## Acknowledgements

U.G.B. is supported by a research training support grant from the EPSRC (EP/N509668/1), with additional equipment support from EP/M028135/1. D.B.P. acknowledges the Royal Academy of Engineering for support. D.P.B. thanks Dr. Stephen Simpson and Luke Debono for advice on hydrodynamics modelling. The authors are grateful to Prof. Miles Padgett for access to the optical tweezers equipment used in the work, and Prof. John Rarity for access to the Nanoscribe direct laser writing system used for micro-rotor fabrication.

## Author contributions

D.B.P. conceived the idea for the project, and developed it with U.G.B and J.M.T. U.G.B. performed the experiments, analysed the data, developed the control loop theory and wrote the control loop software, supervised by J.M.T. and D.B.P. G.M.G. built and maintained the optical tweezers system, and assisted with code modifications. D.B.P., Y.-L.D.H. and M.T. fabricated the laser-written micro-rotors. U.G.B. and D.B.P. performed flow-field simulations. U.G.B., J.M.T. and D.B.P. wrote the manuscript, with input from all other authors.

## Additional information

**Competing interests:** The authors declare no competing interests.

