## [Peer Review File · Nature Communications]

Reviewers' Comments:

Reviewer #1:

Remarks to the Author:

In this manuscript the authors demonstrate hydrodynamic and controlled orientation of microscopic objects using the flow fields generated around optically trapped tools. The method is quite ingenious, and although the experimental challenge is quite considerable, the authors successfully demonstrate a number of controlled manipulations of single and multiple particles. The manuscript is well-written, appropriately referenced and very nicely (indeed, quite beautifully) illustrated. The Supplementary Information adds greatly to the main article, both in the use of video files, and also the mathematical methods for implementing the feedback.

I have only a few minor questions for the authors before recommending publication in Nature Communications:

1. Page 1: the authors claim that hydrodynamic manipulation is "minimally disruptive to biological specimens." I would contend that it is certainly less damaging, but hydrodynamic forces certainly influence (and therefore "disrupt") biological specimens, e.g. the directed neuron growth shown in Wu et al Nature Photon. 6 62 (2012). A slightly different choice of words may be appropriate.
2. From the work presented, only in-plane manipulations appear possible, with the particles at the bottom of the sample cell. Could the authors (briefly comment on)
 - a) whether the particles can be steered in the direction perpendicular to the cell surface?
 - b) can the particles be trapped away from the water-glass interface (perhaps at a liquid-liquid interface?)
 - c) can particles ultimately be hydrodynamically confined in three dimensions using this technique?
3. The authors may like to make reference to the work of Williams et al Nature Communications volume 4, Article number: 2555 (2013), as a prior (although different and distinct) demonstration of particle confinement by optically trapped spheres
4. SI, page 2, should the reference to μ_{tr}^{RT} be to E1 17 in SI Note 2?

Reviewer #2:

Remarks to the Author:

This manuscript describes how holographic optical traps can be used to create hydrodynamic flow fields at the micrometer scale and how the resulting hydrodynamic forces can be used to control the motions of colloidal particles with exquisite precision. This indirect micromanipulation technique provides pin-point control without subjecting the sample to the intense laser irradiation required for optical trapping. This is useful for manipulating samples that are easily damaged or not easily trapped. Both cases are illustrated experimentally. Optical forces previously have been used to create microfluidic pumps, often with the goal of conveying colloidal particles along predefined trajectories. Those earlier studies, however, did not provide dynamic feedback-governed control over individual particles.

The manuscript describes dexterous control over the in-plane motions of one or two particles achieved by directly manipulating either microfabricated rotors or three-particle clusters. The manuscript explains that the number of independent controllable degrees of freedom is limited by the appearance of nodes in the associated flow field. Controlling more than two particles may be challenging. Even so, the demonstrated ability to manipulate pairs of spheres creates interesting research opportunities.

This is an interesting technique and is very well documented. I therefore recommend publication in

Nature Communications.

One statement in the first column of the sixth page should be revised. The discussion of optical bottle beams suggest that these are used to "trap particles of higher index than their surroundings." This, of course, is the default condition for trapping by conventional optical tweezers. Bottle-beam traps are needed for particles that absorb light or back-scatter light strongly. I recommend revising this statement accordingly.

The present implementation is implicitly limited to in-plane motions of particles sedimented onto the lower surface of their container. The presence and role of the confining surface should be stated. Among its other effects, the surface influences the flow field generated by the rotors. The presence of the surface also may influence phenomena that otherwise might be studied in this way. Achieving two-dimensional confinement either requires a thin sample container, or else precludes the use of density-matched samples.

Indirect hydrodynamic control could be generalized to enable three-dimensional manipulation of buoyant samples. This would require rotors that can be rotated selectively around multiple axes. Such three-dimensional rotational control has been demonstrated, for example in Shpaisman H, Ruffner DB, Grier DG. Light-driven three-dimensional rotational motion of dandelion-shaped microparticles, *Applied Physics Letters* 102, 071103 (2013). Some discussion of three-dimensional control might be appropriate.

Response to referees

Reviewer #1 (Remarks to the Author):

In this manuscript the authors demonstrate hydrodynamic and controlled orientation of microscopic objects using the flow fields generated around optically trapped tools. The method is quite ingenious, and although the experimental challenge is quite considerable, the authors successfully demonstrate a number of controlled manipulations of single and multiple particles. The manuscript is well-written, appropriately referenced and very nicely (indeed, quite beautifully) illustrated. The Supplementary Information adds greatly to the main article, both in the use of video files, and also the mathematical methods for implementing the feedback. I have only a few minor questions for the authors before recommending publication in Nature Communications:

We thank the reviewer for their encouraging comments about our work.

1. Page 1: the authors claim that hydrodynamic manipulation is "minimally disruptive to biological specimens." I would contend that it is certainly less damaging, but hydrodynamic forces certainly influence (and therefore "disrupt") biological specimens, e.g. the directed neuron growth shown in Wu et al *Nature Photon.* 6 62 (2012). A slightly different choice of words may be appropriate.

We agree with the reviewer that a different choice of words here would be more suitable. Therefore we have replaced "minimally disruptive" with "minimally damaging".

2. From the work presented, only in-plane manipulations appear possible, with the particles at the bottom of the sample cell. Could the authors (briefly comment on) a) whether the particles can be steered in the direction perpendicular to the cell surface? b) can the particles be trapped away from the water-glass interface (perhaps at a liquid-liquid interface?) c) can particles ultimately be hydrodynamically confined in three dimensions using this technique?

We agree with the reviewer that including more discussion about 3D hydrodynamic manipulation will strengthen our paper. We have added the following paragraph considering the possibility of 3D control on p.8 of the main text:

"Our near-field hydrodynamic manipulation technique may potentially be extended to the third translational dimension, perpendicular to the microscope focal plane. 3D hydrodynamic manipulation of neutrally buoyant target particles (that do not sediment) would be possible by employing optically trapped actuators that can be rotated about arbitrary axes [47, 48], in conjunction with 3D imaging techniques [49]. Generating lift on sedimenting objects would, however, require considerably more powerful flow controllers [44] in order to overcome gravitational forces. For example, far from the substrate the terminal sedimentation velocity v_s of a silica particle of radius $r = 5 \mu\text{m}$ in water is $\sim 60 \mu\text{m/s}$ (estimated by finding the velocity at which gravitational and viscous forces balance [50]). However, we note that $v_s \propto r^2$, and so lift may be more readily generated on smaller particles. Despite these challenges, choreographing the motion of clusters of mobile flow micro-actuators has the potential to yield minimally invasive control over any target degrees-of-freedom that can be measured. We envisage future systems capable of exerting full 3D control over all translational and rota-

tional degrees-of-freedom of mobile cells, using the minimally-invasive forces of water alone."

We think that reviewer's suggestion about trapping at a liquid-liquid interface is intriguing and hope to consider it further in future. However, we believe that it would be a complex and challenging system to analyse and therefore do not wish to consider it in the present paper.

3. The authors may like to make reference to the work of Williams et al Nature Communications volume 4, Article number: 2555 (2013), as a prior (although different and distinct) demonstration of particle confinement by optically trapped spheres

We thank the reviewer for drawing our attention to this work and now reference it on p.6 of the main text with citation [34].

4. SI, page 2, should the reference to μ_{tr}^{RT} be to E1 17 in SI Note 2?

We thank the reviewer for spotting this, and have corrected the mentioned reference to say Eqn. 17 instead of Eqn. 16.

Reviewer #2 (Remarks to the Author):

This manuscript describes how holographic optical traps can be used to create hydrodynamic flow fields at the micrometer scale and how the resulting hydrodynamic forces can be used to control the motions of colloidal particles with exquisite precision. This indirect micromanipulation technique provides pin-point control without subjecting the sample to the intense laser irradiation required for optical trapping. This is useful for manipulating samples that are easily damaged or not easily trapped. Both cases are illustrated experimentally. Optical forces previously have been used to create microfluidic pumps, often with the goal of conveying colloidal particles along predefined trajectories. Those earlier studies, however, did not provide dynamic feedback-governed control over individual particles.

The manuscript describes dexterous control over the in-plane motions of one or two particles achieved by directly manipulating either microfabricated rotors or three-particle clusters. The manuscript explains that the number of independent controllable degrees of freedom is limited by the appearance of nodes in the associated flow field. Controlling more than two particles may be challenging. Even so, the demonstrated ability to manipulate pairs of spheres creates interesting research opportunities. This is an interesting technique and is very well documented. I therefore recommend publication in Nature Communications.

We are grateful to the reviewer for their favourable comments about our work.

One statement in the first column of the sixth page should be revised. The discussion of optical bottle beams suggest that these are used to "trap particles of higher index than their surroundings." This, of course, is the default condition for trapping by conventional optical tweezers. Bottle-beam traps are needed for particles that absorb light or back-scatter light strongly. I recommend revising this statement accordingly.

We thank the reviewer for pointing this out and replace "trap particles of higher index than their surroundings" with "trap absorbing particles and particles of lower index than their surroundings".

The present implementation is implicitly limited to in-plane motions of particles sedimented onto the lower surface of their container. The presence and role of the confining surface should be stated. Among its other effects, the surface influences the flow field generated by the rotors. The presence of the surface also may influence phenomena that otherwise might be studied in this way. Achieving two-dimensional confinement either requires a thin sample container, or else precludes the use of density-matched samples.

We agree with the reviewer that the presence of the confining surface should be stated more clearly in the manuscript. Therefore we have added the following sentence to the section "Hydrodynamic clamping of a single particle" on page 3:

"We note that in all experiments we demonstrate, the target particles are sedimented to the bottom surface of the sample, eliminating the need to hydrodynamically control their motion in the third spatial dimension (z)."

We also highlight that we mention in Supplementary Note 1 that the effect of the presence of a planar surface at the bottom of the sample can be taken into account in the calculation of the flow fields through a modification to the mobility tensor.

Indirect hydrodynamic control could be generalized to enable three-dimensional manipulation of buoyant samples. This would require rotors that can be rotated selectively around multiple axes. Such three-dimensional rotational control has been demonstrated, for example in Shpaisman H, Ruffner DB, Grier DG. Light-driven three-dimensional rotational motion of dandelion-shaped microparticles, Applied Physics Letters 102, 071103 (2013).

Some discussion of three-dimensional control might be appropriate.

We agree with the reviewer here, and have addressed this by including a paragraph in the discussion of the main text, exploring the possibility of 3D hydrodynamic manipulation of neutrally buoyant and sedimenting target objects (details given in our reply to comment 2 of the first reviewer), where we also reference the suggested paper.

We thank the referees again for taking the time to review our work.

Reviewers' Comments:

Reviewer #1:

Remarks to the Author:

The authors have responded appropriately to all points raised in my initial report on the manuscript, and so I am happy to recommend it be accepted.

Reviewer #2:

Remarks to the Author:

The authors have revised their manuscript to address the concerns raised by both reviewers during the first round of review. I consider the result to be suitable for publication in Nature Communications.

Response to referees

Reviewer #1 (Remarks to the Author):

The authors have responded appropriately to all points raised in my initial report on the manuscript, and so I am happy to recommend it be accepted.

Reviewer #2 (Remarks to the Author):

The authors have revised their manuscript to address the concerns raised by both reviewers during the first round of review. I consider the result to be suitable for publication in Nature Communications.